# Assigning the Sex-Specific Markers via Genotyping-by-Sequencing onto the Y Chromosome for a Torrent Frog *Amolops mantzorum*

**DOI:** 10.3390/genes11070727

**Published:** 2020-06-30

**Authors:** Wei Luo, Yun Xia, Bisong Yue, Xiaomao Zeng

**Affiliations:** 1Chengdu Institute of Biology, Chinese Academy of Sciences, Chengdu 610041, China; luowei@cib.ac.cn (W.L.); xiayun@cib.ac.cn (Y.X.); 2Key Laboratory of Bio-Resources and Eco-Environment of Ministry of Education, College of Life Sciences, Sichuan University, Chengdu 610065, China; bsyue@scu.edu.cn

**Keywords:** sex chromosomes, GBS, sex-linked markers, *Amolops mantzorum*

## Abstract

We used a genotyping-by-sequencing (GBS) approach to identify sex-linked markers in a torrent frog (*Amolops mantzorum*), using 21 male and 19 female wild-caught individuals from the same population. A total of 141 putatively sex-linked markers were screened from 1,015,964 GBS-tags via three approaches, respectively based on sex differences in allele frequencies, sex differences in heterozygosity, and sex-limited occurrence. With validations, 69 sex-linked markers were confirmed, all of which point to male heterogamety. The male specificity of eight sex markers was further verified by PCR amplifications, with a large number of additional individuals covering the whole geographic distribution of the species. Y chromosome (No. 5) was microdissected under a light microscope and amplified by whole-genome amplification, and a draft Y genome was assembled. Of the 69 sex-linked markers, 55 could be mapped to the Y chromosome assembly (i.e., 79.7%). Thus, chromosome 5 could be added as a candidate to the chromosomes that are particularly favored for recruitment in sex-determination in frogs. Three sex-linked markers that mapped onto the Y chromosome were aligned to three different promoter regions of the *Rana rugosa*
*CYP19A1* gene, which might be considered as a candidate gene for triggering sex-determination in *A. mantzorum*.

## 1. Introduction

Mammals exhibit XY sex chromosomal systems, and sex-determination is controlled by a conserved male-inducing factor (SRY) on the degenerate Y chromosome [1]. Birds possess ZW sex chromosomal systems, and their sex-determination is accomplished by the dosage of DMRT1 on the Z chromosome [2]. In contrast with the high conservation of mammals and birds, amphibians possess two types of heterogametic systems (XX/XY and ZZ/ZW) for genetic sex-determination, and the sex-determining systems and sex chromosome pairs may vary amongst closely related species, or even amongst populations of the same species [3,4].

Recently, the results from the comparative mapping of sexual differentiation genes have shown multiple origins for the sex chromosomes in anurans [5,6]. Uno et al. (2008) conducted comparative genomic hybridization for *Xenopus laevis* and *Xenopus tropicalis*, and Fluorescence in situ hybridization (FISH) mapping for eight sexual differentiation genes of *X*. *laevis*, *X*. *tropicalis* and *Rana rugosa*. The results revealed that the sex chromosome was independently acquired between *X. laevis* and *X*. *tropicalis*, and that the origins of the sex chromosomes are different between *X. laevis* and *Rana rugose* [5]. In addition, Uno et al. (2015) performed the chromosome mapping of sex-linked genes, and constructed comparative maps of sex chromosome to further confirm the conclusion that the sex chromosomes of *X. laevis*, *X. tropicalis* and *R. rugosa* are different chromosome pairs [6]. Those results seemed to indicate that the origins of sex chromosomes in anurans are extraordinary diverse. Although diversity in the origins of sex chromosomes are found in this clade, those sex chromosomes might be selected from a limited number of candidates, and not randomly chosen from all chromosomes [7]. Indeed, based on a series of allozyme loci, sex-linkage gene and karyotypic analyses of 32 species or populations, six different chromosomes are predicted to be the sex-determining candidates in frogs [8,9].

Such a multitude of origins for the sex chromosomes in anurans suggests that a set of genes might exist, and be preferentially recruited, as the master sex -determination gene. Recent discoveries indicate that a gene that triggers sex-determination is not evolutionarily conserved across vertebrates, and cases of such a gene have already been documented in frogs. According to sex-linkage and karyotypic data, eight genes are predicted to be candidates for determining sex in frogs, including *DMRT1, SOX3, CYP19A1, CYP17, AMH, FOXL2, SF1* and *AR* [9]. *DM-W* is a female-determining gene, and is a truncated copy of *DMRT1* on the W chromosome in *X*. *laevis*. Its homologs have been identified in closely related species, such as *Xenopus Andrei* and *Xenopus clivii*, whereas no *DM-W* orthologs have been found in the distantly related species *X*. *tropicalis*, *Xenopus borealis* and *Xenopus muelleri* [10,11]. These results suggest that the sex-determining gene is not conserved in this lineage. *CYP17* is a candidate female-determining gene on sex chromosome 9 of *Buergeria buergeri*, which is located in the inverted region of the W chromosome; the nucleotide and amino acid sequences of *CYP17* have diverged between Z and W chromosomes [6]. These genes, such as *DMRT1*, *SOX3*, *CYP19A1* and *CYP17*, are the primary components in the sex-determining pathways. These results, taken together, suggest that genes encoding transcription factors and regulating the expression of other genes are particularly suitable for acting as trigger genes for sex-determination. These genes could then be repeatedly recruited as a master gene to govern sex-determination in highly divergent groups [9]. However, those cases are limited to very few species, and the master sex-determining genes, from a limited number of sex chromosome candidates, are yet to be identified.

*Amolops mantzorum*, a torrent frog, is widely distributed in the high mountain streams of southwestern China [12]. Karyotypic studies showed that *A*. *mantzorum* has a male-heterogametic system with heteromorphic sex chromosomes [13], which is rarely observed among amphibians. In this study, we completed Y chromosome sequencing using mechanical microdissection and the single-cell whole-genome amplification technique in *A*. *mantzorum*, and accordingly identified sex-specific markers via genotyping-by-sequencing (GBS) in this species. Next, we assigned those sex-specific markers onto the Y chromosome to further confirm the sex-determination system and sex chromosomes. In addition, we also investigated sex-linkage genes in the sex chromosomes of this frog.

## 2. Material and Methods

A brief description of the methods used in this study is exhibited in Figure 1.

### 2.1. Sampling and Preparation

A total of 197 adults were collected from 15 natural populations (Appendix A). Physiological sex was determined by histology. Muscular tissues were taken from both females and males and stored in 96% ethanol for subsequent analyzing. A total of 40 samples from a single site (Yanjinping, Baoxing, Sichuan, China) were used for RAD-sequencing, including from 21 males and 19 females, while an additional 161 samples from 15 populations were used to PCR verification.

In order to gain the Y chromosome of *A*. *mantzorum*, a single male was collected from Liangluxiang, Tianquan, Sichuan, China. (Appendix A), and the mitotic metaphases were prepared from bone marrow by the air-dry method, as described by Schmid et al. [14]. 

All animal procedures were approved by the Animal Care and Use Committee of Chengdu Institute of Biology (CIB), Chinese Academy of Sciences (Permit Number: CIB-20121220A).

### 2.2. DNA Extraction and Genotyping-by-Sequencing

We extracted genomic DNA from muscular tissues using the Qiagen^®^ DNeasy Blood and Tissue extraction kit (QIAGEN, Valencia, CA, USA) according to the manufacturer’s protocol. DNA purity was checked using the NanoPhotometer^®^ spectrophotometer (IMPLEN, Westlake Village, CA, USA) and the concentration was measured using Qubit^®^ DNA Assay Kit in Qubit^®^ 2.0 Flurometer (Life Technologies, Carlsbad, CA, USA).

Genotyping-by-sequencing (GBS) is based on the high-throughput sequencing of genomic subsets targeted by restriction enzymes, which provides an efficient method for large-scale genotyping. We collected GBS data following the protocol of Elshire et al. (2011) [15]. Briefly, Genomic DNA from individual samples was incubated at 37 °C with *MseI* (New England Biolabs, NEB, Ipswich, MA, USA), T4 DNA ligase (NEB), ATP (NEB) and a *MseI* Y adapter N-containing barcode. Then, restriction-ligation reactions were heat-inactivated at 65 °C. Next, we performed the PCR reaction using purified samples, and PCR products were isolated to retain fragments of approximately 300–350 bp (with indexes and adaptors) from an agarose gel using a Gel Extraction Kit (QIAGEN, Valencia, CA, USA). The resulting library was sequenced on an Illumina NovaSeq 6000 platform using the 150 bp paired-end protocol. These procedures were performed at Novogene Bioinformatics Technology Co., Ltd., Beijing, China (www.novogene.cn).

### 2.3. Filtering and SNP Calling

We used the software package Stacks-2.41 for processing the GBS data because of its high flexibility and its ability to manage large de novo GBS data sets [16]. Firstly, Raw GBS reads were de-multiplexed and filtered using the *process_radtags* algorithm in the Stacks-2.41 program. At this step, reads of low quality, missing the restriction site, were removed. The next stage was to experiment with the pipeline on a representative subset of samples; this allowed us to survey the general properties of the data set and to assess which parameter values are most suitable [17]. At this stage, the main parameters were considered, such as minimum depth of coverage (m), maximum number of mismatches between stacks (alleles) within an individual (M), and maximum number of mismatches between individuals (n). We investigated a series of combinations of those assembly parameters to confirm the best one [a range of M and n values from 1 to 9 (fixing M = n and m = 3)]. Then the Stacks *denovo_map.pl* pipeline was applied to the full data set, using the parameters chosen in the previous stage.

### 2.4. Sex-Specific Locus Discovery and Isolation

Our goal was to isolate single nucleotide polymorphisms (SNPs) and sex-limited markers that are sex-linked in *A*. *mantzorum*. Three approaches were used to identify sex-linked markers for both male (XY) and female (ZW) heterogametic systems [18]. These methods are mainly based on the three sex-marker characteristics that would be expected on the heterogametic sex chromosome. Taking male (XY) heterogametic systems as an example, three approaches are described briefly below.

The first approach screened for SNP loci, with frequency differences based on sex differences in allele frequencies. The output (populations.sumstats.tsv) of the Stacks *populations* module was used to identify such sex-linked loci. In XY systems, sex-linked SNPs should occur in two copies in females, but only one in males. However, considering a few sequencing errors and recombination events, we defined an SNP as sex-linked if one allele had a frequency ≥ 0.95 in females, and a frequency difference ≥ 0.45 between sexes.

The second approach screened for SNP loci based on sex differences in heterozygosity. A locus was considered sex-linked if it is homozygous in half of the females and heterozygous in at least half of the male samples. Such loci were also isolated from populations.sumstats.tsv.

The third approach screened for the sex-limited occurrence loci that are specific to one sex. We isolated sex-limited GBS loci from the *sstacks* output (matches.tsv). In this approach, loci are considered as sex-linked if they are completely absent in female and present in at least half of the males.

In all approaches, the reverse is true for female (ZW) heterogametic systems, and we finally used an R script to identify such putative sex-linked loci from the Stacks outputs [19].

### 2.5. Validation of Sex-Linked Markers

When screening large SNP datasets for sex-linked loci, it is imperative to eliminate false positives. As such, two methods were used to validate putative sex-linked loci. Firstly, we used a Linux grep command to remove the putative sex-limited GBS-tags that occurred in the original read files from the opposite sex. Any sex-limited markers with one or more matches in the raw reads of the opposite sex were excluded from subsequent analyses. Next, all retained loci from the above step were aligned to a female *A*. *mantzorum* draft genome assembly (GenBank Accession No. SRR5248586) [20] and significant alignments were excluded.

Secondly, we used Blastn 2.6.0+ [21] to align the putative sex-linked SNP loci, which were identified with approaches one and two to the female *A*. *mantzorum* draft genome assembly. The locus, that aligned significantly to the female *A. mantzorum* genome assembly and being consistent with Stacks output, would be reserved. All loci that passed this stage were called “confirmed” sex-linked markers.

### 2.6. PCR Verification

In order to validate that the confirmed sex-linked markers were truly sex-linkage, and not specific to the individuals in GBS, PCR and gel electrophoresis assays were designed to amplify the DNA of additional samples. The examples illustrate an XX/XY system, but the process is similar to that of the ZZ/ZW system. We designed two types of primers based on the sequence of confirmed sex-linked markers (Figure 2). Firstly, both of the forward and reverse primers were designed at variable sites of the Y allele, respectively, and the male-specific base was designed as the first base of the 3′ of the primer, to ensure that the primers would restrict PCR amplification to just the Y allele (Figure 2a). We used 3.5 % agarose gel to separate the target sequences with electrophoresis. After separation by gel electrophoresis, we expected to see a band in the male but not the female. Secondly, we designed the primers in a conserved sequence of male and female individuals, to ensure successful amplification in both sex (Figure 2b), and PCR products were sequenced by the Sangon Biotech company, Shanghai, China. After sequenced, we expected to see the heterozygosity in males and the homozygosity in females at variable sites. Primers of these markers were designed with Primer5. For each pair of primers we established the following amplification conditions: an initial 5 min denaturation at 95 °C, followed by 32 cycles at 95 °C for 30 s, 55 °C annealing temperature for 30 s, and 72 °C extension for 30 s, with a final extension of 7 min at 72 °C.

### 2.7. Y Chromosome Assembly and Sex-Linked Markers Assignment

For *A*. *mantzorum*, the subtelocentric Y (chromosome 5) was formed with a large chromosome pericentric inversion [22,23], which could easily be identified with other chromosomes under light microscopy (Figure 1, Y chromosome microdissection). We completed Y chromosome isolation according to the protocol of Zimmer et al. (1997) [24], and DNA amplification following the protocol of Yuan et al. (2017) [25]. Briefly, one male individual (voucher No. xm6395) was chosen for Y chromosome isolation in this study. The Y chromosome dissection was carried out following improved chromosome mechanical microdissection methods [25]. About 10 copies of the desired Y chromosome were successfully microdissected, and the isolated chromosomes were amplified by the popular single-cell amplification method: the GenomePlex^®^ Whole Genome Amplification (WGA) kit (Sigma Chemical Co., Saint Louis, MO, USA). The products of WGA were shipped to Novogene Bioinformatics Technology Co., Ltd., Beijing, China (www.novogene.cn) for library construction, and sequenced with Illumina HiSeq 2500 platform (250 bp paired-end reads). 

Raw Illumina reads were quality-checked with FASTQC (http://www.bioinformatics.babraham.ac.uk/projects/fastqc/), and adaptors were trimmed using Trimmomatic [26]. In this step, raw reads with more than 10% mis-sequenced nucleotides (poly-N) were discarded, and reads for which more than 50% of the bases had a Q-value ≤ 20 were filtered out, and the filtered reads were trimmed to 224 bp nucleotides. We performed a de novo assembly using SOAPdenovo2 for the trimmed reads [27].

Local NCBI-BLAST software 2.6.0+ [21] was used to align the sex-linked locus to the Y chromosome assembly, and the top matching of sex-linked loci were retained if their e-value ≤ 1e-20.

### 2.8. Genes Involved in Sex-Determination or Sex Differentiation on Y Chromosome 

We also identified the potential function of these sex-linked markers assigned on the Y chromosome. These sex-linked GBS loci were directly searched against the NCBI nucleotide database, using Blastn (https://blast.ncbi.nlm.nih.gov/Blast.cgi) to obtain their potential functional information.

## 3. Results

### 3.1. Genotyping-by-Sequencing

We obtained a total of 270,015,340 raw Illumina reads. Raw data were de-multiplexed and filtered using the *process_radtags* module. After the removal of low-quality sequences and reads with missing or ambiguous barcodes, 269,724,145 reads were retained, with an average of 6,743,103 reads per sample. The de-multiplexed and filtered GBS data used in this study have been deposited in the NCBI Sequence Read Archive (Accession no. PRJNA617653).

A subset of representative samples (five males and five females) was chosen for parameter testing (Appendix A). By comparing the results obtained with different parameter combinations, we plotted two statistics: the number of polymorphic loci shared across 80% samples, and the distribution of the number of SNPs per locus (Appendix A). For this data set, M = 2 provided the highest amount of polymorphism across 80% samples (Appendix A), and M = 2 is sufficient to stabilize the proportions of loci with 1–6 SNPs (Appendix A). After a set of parameters had been chosen (M = 2, n = 2, m = 3), the optimal parameters combination were applied to the full data set. We produced a catalogue containing 1,015,964 loci, using the Stacks *denovo_map.pl* pipeline. Of the loci identified, 524,737 were monomorphic and 491,227 were polymorphic.

### 3.2. Sex-Specific Locus Discovery and Isolation

The approach based on frequency differences identified 28 sex-linked SNPs with an XY pattern, which were located on 18 GBS-tags. By contrast, no single marker matched the ZW pattern. The approach based on heterozygosity differences identified 78 sex-linked SNPs with an XY pattern, located on 65 GBS-tags, of which 8 belonged to the set obtained using approach one. We did not find SNPs with a ZW pattern via this approach either. Finally, the approach based on sex-limited occurrence identified 56 male-limited and 10 female-limited GBS-tags. In short, these three methods identified a total of 141 putatively sex-linked markers. All details, putatively of sex-linked markers, are described in Appendix A.

### 3.3. Validation of Sex-Linked Markers

Screening for sex-linkage yielded markers that fit the expectations for either an XY or a ZW system in our GBS datasets, pointing to the occurrence of false positives. We first checked if the putative sex-limited GBS-tags were present in only one sex. On the one hand, we searched for 56 putative male-limited GBS-tags in the female raw reads file, using the Linux grep command, and 32 male-limited tags were retained. On the other hand, 10 female-limited GBS-tags were searched for in the male raw reads, and no female-limited markers were retained. Next, 4 of the 32 putative male-limited GBS-tags had multiple Blast hits in the *A*. *mantzorum* female draft genome assembly, with 99–100% identity. Thus, these four loci were precluded from further analysis. Next, we extracted the assembly of 28 confirmed male-limited GBS loci from *catalog* for subsequently analysis. We then noted that there were four duplicate sequences in these assembled GBS loci (863,235 and 871,854; 871,860 and 882,819). The sequences 869,195 and 883,421 were two members of an assembly. Considering this case, we retained 863,235, 871,860 and 883,421 for subsequently analysis. Finally, 25 out of the 56 male-limited GBS loci were confirmed as sex-linked markers. A total of 75 putative sex-linked SNPs assembly were extracted from *catalog*. Aligning these loci to the draft female *A*. *mantzorum* genome assembly left us with 48 sex-linked SNP loci; putative sex-linked SNPs from approach one and approach two left 4 and 44, respectively. We found that eight assembled loci were duplicate sequences (24,363 and 43,213; 52,538 and 307,750; 63,643 and 199,967; 85,771 and 228,137), but we left 24,363, 52,538, 63,643 and 85,771 for further analysis. Therefore, a total of 44 sex-linked SNP loci was confirmed. Altogether, we detected that a total of 69 confirmed sex-linked loci from the GBS data had an X-Y pattern, and none had a Z-W pattern (the assembled sequence of each locus was shown in Appendix A). All details of confirmed sex-linked markers were described in Appendix A.

### 3.4. PCR Verification

A total of 69 confirmed sex-linked markers have been obtained by the methods presented here. From these, eight sex-linked markers were randomly chosen for PCR verification. Two sex markers were used to design primers following the first type of primer-design method. All of these markers were detected by agarose gel electrophoresis after amplification of 161 additional samples from 15 populations (Appendix A). All individuals exhibited differences between males and females after agarose gel electrophoresis (Figure 3: a total of 18 identical individuals were exhibited for locus 422,993 and 265,333; specimen numbers are shown in Appendix A). Six sex-linked markers were used to design primers following the second type of primer-design method. After the amplification of additional samples (12 male, 12 female) from five populations (Appendix A), the PCR products were sequenced at Sangon Sequencing Center (Shanghai, China). Except for some individuals that were not successfully sequenced, all the male individuals and female individuals showed heterozygosity and homozygosity, respectively, at their polymorphic sites (Appendix A). All details of the primers are shown in Table 1. 

### 3.5. Y Chromosome Assembly and Sex-Linked Markers Assignment

A total of 8,528,986 raw Illumina reads were obtained from sequencing on the Illumina HiSeq 2500 using the PE250 protocol for WGA products. After eliminating low quality reads and adapter sequences, 5,884,104 clean reads were obtained. The de novo assembly of clean data used the SOAPdenovo2 program. It generated 201,305 contigs, with an average length of 243 bp and an N50 of 250 bp. Of the 69 confirmed sex-linked GBS-tags, 55 could be mapped on the *A*. *mantzorum* Y chromosome (Appendix A). Sequencing reads were submitted to the National Center for Biotechnology Information’s (NCBI) Short Read Archive (Accession no. PRJNA622419).

### 3.6. Identifying Potential Sex-Determining Genes

In order to understand the potential function of these sex-linkage markers assigned on the Y chromosome, we directly searched against the NCBI nucleotide database, using Blastn, and three sex-linkage GBS markers were mapped onto three different regions of *R. rugosa*’s CYP19A1 gene (Table 2).

## 4. Discussion

### 4.1. Chromosome 5 is the Sex Chromosome Pair

Previous cytogenetic studies have demonstrated that the *A. mantzorum* species, having the diploid chromosome number of 26, presents an XY sex-determination system [13]. The chromosome heteromorphism was found in the chromosome pair 5 in males, which is composed of a subtelocentric chromosome (ST) and a metacentric chromosome (M), as can be easily identified under light microscopy. The 5S rDNA was detected in the telomeric regions of the short arms of both the X and Y homologues, clearly demonstrating that the subtelocentric Y could be reconstructed from the X chromosome by pericentric inversions [23]. The C-positive heterochromatins were observed in the middle regions of the long arm in the Y chromosome, indicating that these two originally homologous chromosomes start to differentiate [28].

In the present study, a total of 69 sex-linked sequences, identified by genotyping-by-sequencing, showed a male heterogamety, verified by PCR-based analysis with a large number of samples covering the whole geographic region of the species, which is consistent with the previous cytogenetic investigation of *A. mantzorum*. In order to further identify the role of chromosome 5 in this species, the Y chromosome was dissected under light microscopy and subsequently the amplification was performed, followed by sequencing and assembly. Then, those sex-linked markers via GBS were taken and mapped onto the Y chromosome by homologous sequences matching, using the BLAST search, with 55 markers successfully mapped out of 69 (i.e., 79.7%), providing high confidence regarding the identity of the sex chromosomes in the species. Our results give strong evidence that the population of *A. mantzorum* under study has a strictly genetic sex-determination system, with male heterogamety (supported by 69 informative GBS-tags). According to the cytogenetic analyses, chromosome 5 is the sex chromosome pair, which possesses a set of sex-linked loci.

### 4.2. Particular Chromosomes Make Better Sex Chromosomes

In sharp contrast with mammals and birds, the sex chromosomes in amphibians are mostly homomorphic. One possible explanation for this phenomenon is the high rate of sex chromosome turnover [29,30]. Indeed, frequent turnovers of sex chromosome have been reported in amphibians [3,31]. However, comparing known sex chromosomes and genes across vertebrate lineages reveals that some chromosomes and genes are repeatedly recruited as master regulators for sex-determination in distantly related animals [7]. Graves and Peichel (2010) proposed the “limited options” hypothesis, suggesting that a single set of genes/chromosomes are more likely to be recurrently recruited for sex-determination than others, and are preferentially reused over vertebrates [7]. In other words, although rapid turnovers of sex chromosomes and genes had already occurred in many vertebrate lineages [30], these turnovers are not random. Chromosomes bear the genes that are extraordinarily suited to a sex-determination role, or acquire a new sex-antagonistic mutation locus, such as DMRT1, which would be repeatedly selected as the sex chromosome [7]. 

In fact, such cases actually have already been documented in frogs [8,9]. According to an examination of sex-linkage and karyotypic data, sex chromosomes are repeat turned over among chromosomes 1, 2, 3, 4, 7 and 9 in frogs, and eight genes that are located on these six chromosomes are particularly favored for recruitment as sex-determining candidates in this clade, such as *DMRT1*, *SOX3*, *CYP19A1* and *AR* [9]. The frog *A. mantzorum*, which had 2n = 26 chromosomes with five large and eight small pairs, belongs to the *Amolops* genus of Ranidae [32]. Of great interest, cytological research suggests that the chromosome pair 5 is the sex chromosome pair in this species. In this study, our sex-linked molecular markers also confirmed the above results.

In conclusion, our results add the Sichuan torrent frog to the list of frogs that display male heterogamety, and provide molecular markers for further exploration of the Y chromosome in this species. Thus, together with chromosome 5, seven chromosomes, including 1, 2, 3, 4, 5, 7, and 9, are particularly favored for recruitment for sex-determination over others in frogs.

### 4.3. Sex-Determination and Differentiation Genes Involved in A. Mantzorum

At present, we have confirmed that the chromosome pair 5 is the sex chromosome pair in *A. mantzorum*. However, none of the genes that participated in sex-determination have been found in this species. Under this study, those sex-linked markers that were identified by GBS can provide an entry point to identifying sex-linkage sequences from sex chromosomes assemblies, thereby expanding our knowledge of the genes that are located on the sex chromosome in *A. mantzorum*; if the sex markers lie close to the genes, we may find the genes that are involved in sex-determination and differentiation.

In our study, a total of three hits were found for the 55 confirmed sex-linked markers that were mapped onto the sex chromosome through the blast search against the NCBI nucleotide database. Specifically, three sex-linkage markers (174,608; 201,126; 408,566) were located on *R. rugosa*’s *CYP19A1* gene. To our knowledge, *CYP19A1* is highly conserved in vertebrates as the sexual differentiation gene, and this gene codes Cytochrome P450 aromatase, which is the estrogen synthesizing enzyme and plays a crucial role in the gonadal differentiation and development [33,34,35]. At the present time, the dimorphic expression of *CYP19A1* has been found in several anuran species, such as in *R*. *rugosa* and *X*. *laevis* [36,37,38]. In these species, the level of *CYP19A1* expression was significantly increased during ovarian differentiation, indicating that this gene is closely implicated in ovarian differentiation in frogs. Altogether, these results imply that *CYP19A1* is an important component of the sex-determining cascade in anuran species. Detecting the expression of *CYP19A1* in embryonic gonads, in species with temperature-dependent sex-determination (TSD), occurs at female-producing temperature, while it is almost undetectable at a male-producing temperature [39,40,41]. High temperature-induced masculinization is accompanied by an increase in gonadal DNA methylation at the *CYP19A1* promoter, and the subsequent suppression of *CYP19A1* expression [41,42]. Matsumoto et al. (2016) suggested that DNA hypomethylation and H3K4me3 modification at the *CYP19A1* promoter initiate a cascade of ovarian differentiation in red-eared slider turtles (*Trachemys scripta*) [41]. These findings suggested that the dimorphic expression of *CYP19A1* may be a crucial role in determining gonad sex trajectories, even in species with TSD. Interestingly, three sex-linked markers (Y-specific SNP), located on *A. mantzorum*’s Y chromosome, were mapped onto three different regions of *R. rugosa*’s *CYP19A1* gene. From these results, taken together, we speculate that the nucleotide sequence divergence of this gene, between the X and Y chromosomes, had already occurred, and that this gene may also be involved in sexual differentiation and development in *A. mantzorum*. The *CYP19A1* in *A. mantzorum* represents similarities with the *AR* (androgen receptor) gene in *R. rugosa*, which has ZZ/ZW sex chromosomes. The *AR* gene is located on the sex chromosome in *R. rugose* [5], like the *CYP19A1* gene in *A. mantzorum*. The divergence of nucleotide sequences had occurred in the promoter regions of both Z- W-*AR*. In addition, the expression of *AR* in the ZZ-males was two times higher than in ZW-females at the sex-determining stage. Thus, the *AR* gene has been considered to be a candidate gene for sex-determination in *R. rugose* [43,44]. Similarly, the *CYP19A1* gene might be considered as a candidate sex-determining gene in *A. mantzorum* frogs.

## Figures and Tables

**Figure 1 genes-11-00727-f001:**
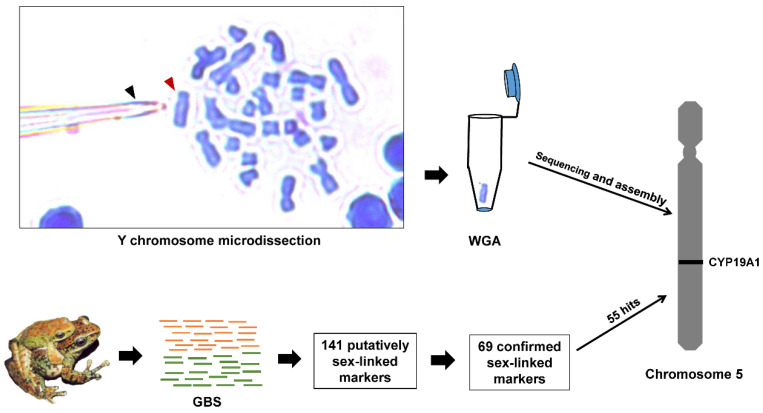
Summary of approaches for assigning sex-linked markers to the Y chromosome of *A. mantzorum*. Y chromosome microdissection: A metaphase spread of *A. mantzorum* chromosomes stained with Giemsa is placed on a slide. The chromosome dissection was carried out following improved chromosome mechanical microdissection methods. The slides were inversely mounted on a glass platform for chromosome collection from below, with chromosomes facing down. We used a glass capillary (black arrow) controlled by a micromanipulator (MMO-203, Narishige Group, Tokyo, Japan) to scrape the Y chromosome (red arrow). The desired Y chromosome was microdissected and collected with a microscope (Leica DM2500, Leica Microsystems GmbH, Wetzlar, Germany) under a water immersion objective lens (63× magnification). WGA: about 10 copies of the sex chromosome were catapulted into a cap tube, and specific DNA of the isolated chromosomes were amplified by the single-cell amplification method, using the GenomePlex ^®^Whole Genome Amplification (WGA4) kit (Sigma Chemical Co., Saint Louis, MO, USA) following the operation manual. Next, we completed the Y chromosome sequencing and assemblage. GBS: A total of 19 females and 21 males were sequenced via genotyping-by-sequencing, and 141 putatively sex-linked GBS-tags were found via three strategies. After excluding false positives, 69 sex-linked GBS-tags were confirmed, and all of the retained GBS-tags pointed to an XY sex-determining system. Sex-linked markers assignment: 55 sex-linked GBS-tags were mapped onto the Y chromosome. These GBS-tags were directly searched for in the NCBI nucleotide database, and 3 GBS-tags were aligned to the *R. rugosa CYP19A1* gene.

**Figure 2 genes-11-00727-f002:**
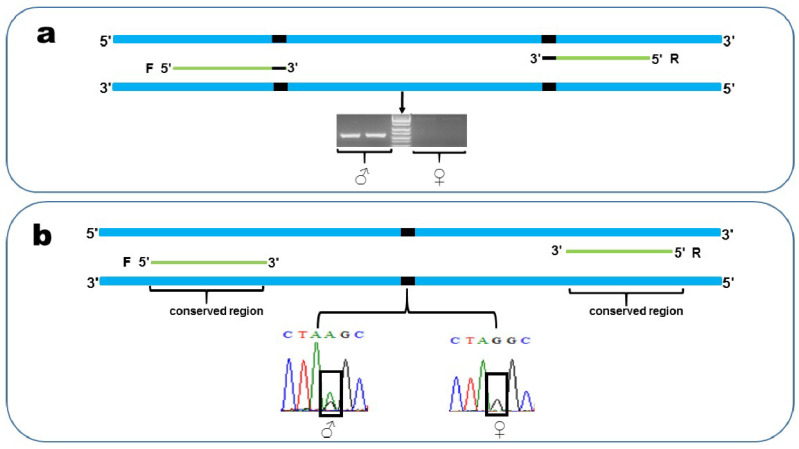
Primer design for sex-linked markers. Examples suggest an XX/XY sex chromosome system, but the results are similar in species with a ZZ/ZW system. All primers (green line) are designed based on the Y alleles (blue line), and ‘F’ and ‘R’ indicate forward primer and reverse primer, respectively. Black segments indicate Y-specific SNPs that do not occur on the X. Female individuals are represented by the symbol ‘♀’ and male individuals are represented by the symbol ‘♂’. (**a**) Both the forward and reverse primers were designed at Y-specific SNPs. The Y-specific base was set as the first base of the 3′ of the primer sequence. With the PCR products separated by gel electrophoresis, we expected to see a band in the male but not the female. (**b**) Both of the forward and reverse primers were designed at the conserved region. After being sequenced, we expected to see heterozygosity in the male and homozygosity in the female at SNPs (black box).

**Figure 3 genes-11-00727-f003:**
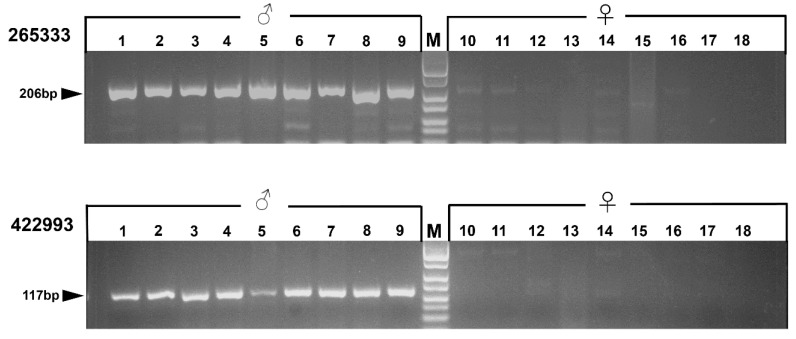
Gel electrophoresis showing the sex-specific PCR amplification of markers 265,333 and 422,993. The locus ID is indicated to the left. Female individuals are represented by the symbol ‘♀’ and male individuals are represented by the symbol ‘♂’. ‘M’ indicates a DNA marker. Black arrows indicate the PCR products size.

**Table 1 genes-11-00727-t001:** Primer sequences of eight sex-linked markers isolated from *Amolops mantzorum*.

Locus	Primer sequence (5′–3′)	T (°C)	Length (bp)
**265333 ***	Forward: GTGCTGCCTCTCGCTTCCCGAT	55	117
	Reverse: AAATGGTCTAAGCTCGGGGTCG		
**422993 ***	Forward: TAAAGGCACGGTGTTCG	55	206
	Reverse: TTCCTCCTGACATAAGAGCTA		
**3662**	Forward: TCGACGTGAGCCTTAGTCAT	55	258
	Reverse: AAAGCAATTCAAACGAGCAT		
**37926**	Forward: TTGATGGTGAATTGATTGGG	55	200
	Reverse: TCACAAATTCCAGGTGCTTG		
**52538**	Forward: CTCCTGGCAACTCCTTTCTG	55	189
	Reverse: GAGCCCATATTCAATTCACC		
**116241**	Forward: GTGTCAGCACAAGAGGTAGG	55	214
	Reverse: ATCTGGTATCATCCGAGAAA		
**144173**	Forward: AAAGCAATTCAAACGAGCAT	55	256
	Reverse: CGACGTGAGCCTTAGTCATAG		
**181302**	Forward: GGGACAGAGTGAGGCTCGCTAA	55	160
	Reverse: CTGAGGATATGCAATCCCGTTG		

* indicates that both of the forward and reverse primers were designed at Y-specific SNPs, and the others were designed at conserved region. T indicates the annealing temperature of PCR thermal cycling. Length indicates PCR product size.

**Table 2 genes-11-00727-t002:** Blast results of nonredundant nucleotides from the NCBI database.

Locus	Length (bp)	NCBI Hit	Query Cover	E-Value	Identity	Start	End
174,608	283	*Rana rugosa CYP19A1* gene for cytochrome P-450 aromatase, promoter region and partial cds (AB379847.1)	97%	2e−49	81.72%	12,255	12,505
201,126	280		95%	2e−53	81.82%	11,730	11,997
408,566	299		21%	2e−13	92.06%	14,151	14,212

The table shows the matched nucleotide alignments from the blast (Blastn) hit results for all 55 sex-linked GBS-tags that mapped onto the Y chromosome. Three sex-linked GBS-tags aligned to completely different position of *R*. *rugosa*’s *CYP19A1* gene, and shared > 81% identity. The Genbank accession number of the hit is shown in parentheses. ‘start’ and ‘end’ indicate the nucleotide position of the *CYP19A1* gene.

## Data Availability

Raw Illumina GBS reads and sex chromosome sequencing reads have been deposited at NCBI Sequence Read Archive, BioProject PRJNA617653 and PRJNA622419. Sex chromosome assembled fasta file, Stacks outputs, PCR sequence and R intermediate files have been archived on Dryad, https://doi.org/10.5061/dryad.mkkwh70wb.

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
