# Peer review of "Assigning the Sex-Specific Markers via Genotyping-by-Sequencing onto the Y Chromosome for a Torrent Frog Amolops mantzorum"

_genes, 2020, doi:10.3390/genes11070727_

Round 1

Reviewer 1 Report

A very well presented topic of sex-determinism in frog, with insertions of modern cytogenetics and molecular genetics studies. 

Reviewer 2 Report

Sex determination system is greatly variable in vertebrates. Along with typical male (XY) or female (ZW) heterogametic system, unusual multiple sex chromosomes (platypus, some lizards, or even monkey) or single X (mole voles) were described. In any system, the main trigger, master switch gene, should evolve, as Sry in mammals. Unless several omissions were known (the same mole vole or spiny rats in placentals), such exception just proved the rule - the master gene should start the genetic pathway for sex determination. For that reason, an allocating the sex-specific markers, which was made by Luo et al., is exciting and relevant. The paper is a continuation of several authors' studies on chromosomal evolution and genome sequencing in Anura. Unless the difference in sex-linked and sex-determining genes exists, the disclosing sex chromosomes content is the first step to discover the master switch gene.

The manuscript is straightforward, and it was a pleasure to read. The methodology is appropriate, the authors studied huge material, and the results are convincing. A CYP19A1 gene, which really might be considered as a candidate gene to trigger sex determination in A. mantzorum, is under intense research in the so-called temperature-dependent sex determination system, for example, in red-eared slider turtles. Analysis of these studies could significantly improve the discussion:

Matsumoto, Y.; Buemio, A.; Chu, R.; Vafaee, M.; Crews, D. Epigenetic control of gonadal aromatase (cyp19a1) in temperature-dependent sex determination of red-eared slider turtles. PLoS ONE 2013, 8, e63599.

Matsumoto, Y.; Hannigan, B.; Crews, D. Temperature shift alters DNA methylation and histone modification patterns in gonadal aromatase (cyp19a1) gene in species with temperature-dependent sex determination. PLoS ONE 2016, 11, e0167362.

Dong, J., Xiong, L., Ding, H., Jiang, H., Zan, J. and Nie, L., 2020. Characterization of DNA methylation and transcript abundance of sex-related genes during temperature-dependent sex determination in Mauremys reevesii. Biology of Reproduction. Volume 102, Issue 1, January 2020, Pages 27–37.

There are also some minor flaws.

Line 12 (abstract) GBS - The abbreviation should be given in full on first use and in keywords (Line 24)

Figure 1, a part of Y chromosome microdissection should be improved: the pattern is blurred; it needs to be made clearer.

Lines 94-95 Latin names should be italicized.

Lines 98-99  A metaphase spread of A. mantzorum chromosomes stained with Giemsa on a slide. - unclear, re-phrase, please.
